# Fermented Feed Supplement Relieves Caecal Microbiota Dysbiosis and Kidney Injury Caused by High-Protein Diet in the Development of Gosling Gout

**DOI:** 10.3390/ani10112139

**Published:** 2020-11-17

**Authors:** Yumeng Xi, Yuanpi Huang, Ying Li, Junshu Yan, Zhendan Shi

**Affiliations:** 1Jiangsu Key Laboratory for Food Quality and Safety-State Key Laboratory Cultivation Base of the Ministry of Science and Technology, Animal Husbandry Institute, Jiangsu Academy of Agricultural Sciences, 50 Zhongling Street, Nanjing 210014, China; xiyumeng@jaas.ac.cn (Y.X.); hyp5360@163.com (Y.H.); zdshi@jaas.ac.cn (Z.S.); 2Animal Diseases Control & Prevention Centre of Jurong City, 86 Renmin Road, Zhenjiang 212400, China; liying_mlsty@163.com

**Keywords:** fermented feed, gosling gout, high-protein diet, kidney injury, caecal microbiota dysbiosis

## Abstract

**Simple Summary:**

Recently, high-protein complete feed has been increasingly used in the Chinese goose industry, increasing the outbreaks of gout in goslings. We found that high-protein diets are implicated in the kidney injury and gut microbiota dysbiosis associated with the occurrence of gout in goslings. Fermented feed, containing many helpful intestinal probiotics and metabolites for birds, shows potential for the protection of gut microbiota. We found that fermented feed supplement alleviates the gout occurrence caused by the high-protein diets and improves renal function, because of its regulations of caecal microbiota. Fermented feed increased *Lactobacillus* and decreased *Enterococcus* in the cecum of goslings. We suggest that goose farmers should strictly control the proportion of protein in gosling feed and consider using fermented feed as an effective measure to control the outbreak of gosling gout.

**Abstract:**

Firstly, forty-eight 1-day-old goslings were randomly allocated to four groups and were fed diets containing crude protein (CP) at different concentrations: 160, 180, 200, and 220 g/kg in Experiment One. We found a dose-dependent relationship between the dietary protein levels and morbidity of gosling gout. The concentration of serum uric acid (UA), creatinine (Cr), and urea nitrogen (UN), and the activity of xanthine oxidase in the 220CP groups were significantly higher than those in the low-protein diet groups. Beneficial microbes, including *Akkermansia*, *Lactococcus*, and *Butyricicoccus* were enriched in the ceca of healthy goslings, while the microbes *Enterococcus*, Enterobacteriaceae, and *Bacteroides* were enriched in those with gout. Then, we explored the effects of fermented feed on gosling gout caused by high-protein diets in Experiment Two. A total of 720 1-day-old goslings were randomly allotted to four experimental groups: CN (162.9 g/kg CP), CNF (167.5 g/kg CP, replacing 50 g/kg of the basal diet with fermented feed), HP (229.7 g/kg CP, a high-protein diet), and HPF (230.7 g/kg CP, replacing 50 g/kg of the high-protein diet with fermented feed). We found that the cumulative incidence of gout increased in the HP group compared with that in the control, but decreased in the HPF group compared to that in the HP group. Similarly, the concentration of serum UA in the HP group was higher than that in the CN group, but decreased in the HPF group. Meanwhile, compared with the HP group, using fermented feed in diets decreased the abundance of *Enterococcus* in the ceca of goslings, while increasing the abundance of *Lactobacillus*. These results suggest that appropriate dietary protein levels and the fermented feed supplement might relieve the kidney injury and gut microbiota dysbiosis caused by high-protein diets in the development of gosling gout.

## 1. Introduction

During recent years, the modernization of the goose industry in China has resulted in a growth rate-oriented production that primarily is based on feeding the geese high-protein complete feeds, instead of the traditional herbaceous roughages, but also leads to physical problems for the geese. Since 2016, the Chinese goose industry has experienced severe widespread outbreaks of gosling gout that have resulted in serious gosling mortality and economic losses [1,2]. Gosling gout is a purine metabolic disorder hyperuricemia that is characterized by an elevated serum uric acid (UA) level and the deposition of urate in and around the internal organs [3,4]. Uric acid is the major product of purine nucleotide metabolism in birds, much of which might originate from the diets. When fed high-protein diets, a large amount of purine nucleotides is absorbed through the intestine and converted into an excess of UA. If it exceeds the removal capacity of the kidney and damages renal functions, the UA can build up in the blood culminating in visceral urate exudation [3,5]. It was reported that increasing the feed crude protein (CP) level to > 280 g/kg or fishmeal level to > 80 g/kg in broiler diets can significantly lower the susceptibility window of visceral gout to a much younger age, such as from days 3–10 [6]. However, there is currently little information on the consequences of the use of high-protein diets on renal function and the occurrence of gout in goslings. It is necessary to evaluate this because gosling gout has brought serious harm and huge economic losses to the goose industry.

In addition to significantly raising the levels of blood UA, high-protein diets usually lead to gut dysbiosis [7]. The modernization of the goose industry has resulted in a growth rate-oriented production that is primarily based on feeding the geese high-protein complete feeds, instead of the traditional herbaceous roughage [8,9], and this is suspected to have a negative influence on gut microbiota. Usually, increased protein ingestion results in an increased transfer of proteins from the small to the large intestine, impacting the compositions of large intestine contents, epithelial cell morphology, and bacterial metabolism [10], which might cause cumulative and chronic toxicity to the intestines [11]. The gut microbiota, likely through the products of microbial metabolism and the modulation of systemic or mucosal immunity, communicates with the kidney and has the potential to affect the prognosis of chronic nephritis [12], kidney stones [13], or gout in humans [14]. In our previous study, we found that gut dysbiosis in goslings might cause the translocation of gut-derived lipopolysaccharide (LPS), or even bacteria, and thereby interferes with kidney functions and causes inflammatory responses and UA transporter disorders [7]. Therefore, gut dysbiosis due to high-protein complete feeds in modern production may play a key role in the development of gosling gout, and protecting the gut microbiota might be an alternative strategy in the control and prevention of gout in goose production.

The fermentation process has been employed to produce functional feeds that have the potential to improve avian gastrointestinal functions and overall health [15]. Compared to raw materials, fermented products tend to have higher number of lactic acid bacteria (LAB), lower number of *Enterobacteriaceae*, higher concentrations of organic acids (mainly lactic acid), and lower pH values [16]. It has been reported that 50–75 g/kg of fermented feed supplement can be particularly beneficial for healthy gastrointestinal microecology and efficient growth performance of geese [15]. Moreover, fermented feed has been shown to decrease broiler mortality rates by positively affecting their immune responses [17]. This feeding strategy not only increases the mucosal immunity, but also induces circulating antibodies in chickens. Fermented feed supplementation at a level of 40 g/kg in chickens, significantly increased plasma IgG and S-IgA (secretory IgA) significantly [18,19]. Besides that, some fermented products showed their potential by even protecting the health of other organs. For example, a fermented extract can be used in the treatment of obesity and non-alcoholic fatty liver disease by modulating the plasma lipids and liver enzymes through the regulation of adipogenic/lipogenic transcriptional factors [20]. Thus, the fermented feed supplement may have beneficial effects in protecting kidney health by regulating the gut microbiota imbalance, and preventing the occurrence of gosling gout. Therefore, the present study aimed to (1) evaluate the degrees of damage to the kidney and caecal microbiota in goslings with different dietary protein levels, and (2) discuss the effects of fermented feed in relieving gosling gout caused by the high-protein diets.

## 2. Material and methods

### 2.1. Ethical Approvals

The study was approved by the Research Committee of Jiangsu Academy of Agricultural Sciences and conducted according to the Regulations for the Administration of Affairs Concerning Experimental Animals (Order No. 63 of the Jiangsu Academy of Agricultural Science on 8 July 2014). The experiments were conducted in accordance with the ARRIVE (Animal Research: Reporting of In Vivo Experiments) guidelines.

### 2.2. Experiment One

Experiment One was conducted from November 2 to 17, 2018 at the Experimental Animal Center in the Jiangsu Academy of Agricultural Sciences (Nanjing, China). Forty-eight 1-day-old Yangzhou geese (*Anser domestica*), supplied by the Tianzhijiao Breeding Geese Limited Company (Chuzhou, China), were randomly allotted to 4 experimental groups (50%♀:50%♂) and fed diets containing different concentrations of CP: 160 g/kg (control, 160CP group), 180 g/kg (180CP group), 200 g/kg (200CP group), or 220 g/kg (220CP group). Although the National Research Council (NRC, 1994) suggests 200 g/kg CP as a dietary protein requirement for 0–3-week-old goslings, Chinese geese have a lower dietary protein requirement than that of European geese. This is because Chinese native breeds usually have a smaller body size and slower growth rate than those of European meat or liver breeds of large heavy geese because of different approaches to and degrees of breeding selection. All geese were raised in a controlled environment house with four stainless steel cages (C606, Petfun, China) of identical size (1.20 × 1.00 × 0.50 m^3^, 12 goslings per cage), which were placed 0.50 m above the ground. The ambient temperature was maintained at 31 °C from days 0 to 3, 29 °C from days 4 to 6, and 28 °C from day 7 until the end of the experiment. The light schedule was as follows: 22 h of light from days 0 to 3, and 18 h of light from days 4 to 15. The relative humidity was approximately 60% throughout the experiment. The experimental period was 15 days.

Goslings had free access to water and feed. The composition of the experimental diets is presented in Table 1. Diet samples (> 1% of fresh feed) were randomly sampled from five locations weekly and mixed for analysis. The daily feed intake and uneaten were measured (the residual amount of feed > 5%) from 6 to 14 d of age, and the body weight (live weight) of all geese was recorded using an electronic scale (002, K-Fine, China) before feeding on days 3, 10, and 15 of the experiment (Appendix A). Twenty-four goslings (n = 6 goslings/treatment, ♀: ♂ = 1:1) were randomly chosen for immediate blood collection (5 mL/gosling) using blood-collection tubes without anticoagulant after decapitation on day 10 and the others were chosen on day 15 for the follow-up tests. The kidney tissues (volume = 1 cm^3^) and duodenum (length = 1 cm) of 24 goslings (n = 6 goslings/treatment) were collected after blood sample collection on day 15 for histological analysis, and their cecum contents were collected in 2 mL sterile internally threaded cryogenic vials (126280, Geriner, Germany) and immediately stored in liquid nitrogen for 16S rDNA analysis. In addition, as one of the selection standards, when the goslings had gross kidney lesions and the serum UA concentrations were above the marginal value of urate supersaturation (male, 416 μmol/L and female, 357 μmol/L) [7,21], they were noted as having gout. The typical predominant gross lesions of kidneys were pale, mottled, and swollen, and the renal tubules and ureters were distended with excess urates. The cumulative morbidity of gout in this study was summed after the experimental period.

### 2.3. Experiment Two

Experiment Two was conducted from November 1 to December 5, 2019 on Mali Ecological Farm in Zhenjiang (Jiangsu Province, China). A total of 720 1-day-old SanHua geese (*Anser domestica*), supplied by the Tianzhijiao Breeding Geese Limited Company (Chuzhou, China), were randomly allotted to four experimental groups (all male), including: CN (162.9 g/kg CP, a basal diet), CNF (167.5 g/kg CP, replacing 5.0% basal diet with fermented feed), HP (229.7 g/kg CP, a high-protein diet), and HPF (230.7 g/kg CP, replacing 5.0% high-protein diet with fermented feed) groups. Each treatment had six replicates, and each replicate had 30 birds. All goslings were raised in a thermostatic house with double-deck stainless steel cages of identical size (1.20 × 1.00 × 0.50 m^3^, 15 goslings per cage), and the lower-deck cages were placed 0.50 m above the ground. The animal management, ambient temperature, relative humidity, lighting, and prophylactic measures were the same as in Experiment One. The composition of the experimental diets is presented in Table 2. For the production of feed fermentation, a seed sourdough was prepared with fermentation substrate (a mixture of corn, soybean meal, and bran with a certain proportion) and multiple-strain culture (*Lactobacillus plantarum*, *Bacillus subtilis*, *Saccharomyces*, and so on). The seed sourdough was used to inoculate feed fermentation, and the fermentation material needed to be stacked to 60–80 cm high in the fermenter (Baohui, China). During fermentation, the fermentation material needed to be stirred, ventilated, and refrigerated, when the temperature of the fermented material in the fermenter was greater than 45 ℃. After 72–120 h of fermentation, the pH of fermentation material reached 3.8–4.2, denoting the completion of fermentation. The ME (metabolizable energy) was calculated. The CP was determined using the Kjeldahl system (Kjeltec 8400 Analyzer Unit, FOSS, Höganäs, Sweden) as total N × 6.25. The crude fiber (CF) was determined using the Ankom 220 Fiber Analyzer Unit (ANKOM Technology Corporation; Macedon, NY, USA). Crude fat was determined by the ether extract method using the Foss Tecator Soxtec 2050 Avanti (FOSS, Denmark). The calcium and phosphorus concentrations in the feed were determined by inductively coupled plasma mass spectrometry (OPTIMA 2100DV; PerkinElmer Co., Waltham, MA, USA). The daily feed offered and refused were measured (the residual amount of feed > 5%) every day, and the body weight (live weight) of all geese was recorded using an electronic scale before feeding every three days during the experiment (Appendix A). Meanwhile, blood samples and kidney and caecal (length = 1 cm) tissues from 24 goslings (n = 6 animals/treatment, 1 gosling in each replicate) were collected on day 14 for serum indices and histological analysis, and their caecal contents were collected in 2 mL sterile internally threaded cryogenic vials and immediately stored in liquid nitrogen for 16S rDNA analysis. The collection methods of these samples were the same as in Experiment One.

### 2.4. Serum Metabolite Measurement

All blood samples were incubated at 37 °C for 2 h after collection, and then centrifuged at 1500× *g* for 15 min. The serum obtained was stored in 0.6 mL tubes (Eppendorf, Germany) at −80 °C for further analysis (Table 3 and Table 4). The serum UA level was determined by phosphotungstic acid colorimetry. The concentration of creatinine (Cr) and urea nitrogen (UN), and the activity of xanthine oxidase (XOD) were determined by enzymatic colorimetry using a microplate spectrophotometer (Promega Corporation, Madison, WI, USA). These kits were supplied by the Jiancheng Bioengineering Institute (Nanjing, China); the codes were C012 (UA), C011-2 (Cr), C013-2 (UN), and A002 (XOD). All assays were performed according to the instructions of the manufacturer. The serum samples were tested in triplicate. The intra-assay and inter-assay coefficients of variation for the assays were <10% and <15%, respectively.

### 2.5. Histomorphological Observation

The renal and intestinal tissue samples collected from goslings were fixed in 4% paraformaldehyde, embedded in paraffin, and sectioned (slice thickness: 3 μm; four slices per gosling). The pathological changes in the kidney and intestinal canal (approximately 7 cm distal to the pyloric sphincter) of goslings were observed under a light microscope after hematoxylin and eosin (HE) staining. The villus height and crypt depth of the intestinal canal were measured with ImageJ (version 1.8.0; National Institute of Health, USA). Eight measurements of different intact villi per slice were recorded (eight measurements in three successive vision fields). Statistical analyses were performed based on the average of 32 measurements per gosling.

### 2.6. 16S rRNA Sequencing of the Cecum Contents

DNA from the cecum content samples in the two experiments was extracted using the MicroElute Genomic DNA Kit (D3096-01, Omega, Inc., USA) following the instructions of the manufacturer. Sample blanks consisting of unused swabs were processed through DNA extraction, and they were checked not to produce 16S amplicon. The total DNA was eluted in 50 µL of elution buffer by a modified procedure described by the manufacturer (QIAGEN) and stored at −80 °C.

Using the total DNA of samples as a template and the 16S rDNA primers (338F 5′-ACTCCTACGGGAGGCAGCAG-3′; 806R 5′-GGACTACHVGGGTWTCTAAT-3′), the V3–V4 region of the bacterial 16S rRNA was amplified. All reactions were carried out in a 25 µL (total volume) mixture containing 25 ng of genomic DNA extract, 12.5 µL of PCR Premix, 2.5 µL of each primer, and PCR-grade water to adjust the volume. The PCR products were normalized by AxyPrep Mag PCR Normalizer (Axygen Biosciences, Union City, CA, USA), which enabled to skip the quantification step regardless of the PCR volume submitted for sequencing. The amplicon pools were prepared for sequencing using the AMPure XT beads (Beckman Coulter Genomics, Danvers, MA, USA) (performed by LC-Bio Technology Co., Ltd, Hangzhou, Zhejiang Province, China). The size and quantity of the amplicon library were assessed using LabChip GX (Perkin Elmer, Waltham, MA, USA) and the Library Quantification Kit for Illumina (Kapa Biosciences, Woburn, MA, USA), respectively. The PhiX Control library (v3) (Illumina) was combined with the amplicon library (expected at 30%). The library was clustered to a density of approximately 570 K/mm^2^. The libraries were sequenced either on 300PE MiSeq runs, and one library was sequenced with both the protocols using the standard Illumina sequencing primers, eliminating the need for a third (or fourth) index read.

The reads were filtered using the Quantitative Insights Into Microbial Ecology (QIIME; http://qiime.org/tutorials/processing_illumina_data.html) quality filters. The CD-HIT pipeline was used to pick operational taxonomic units (OTUs) by preparing an OTU table. The sequences were assigned to the OTUs at 97% similarity. The representative sequences were chosen for each OTU, and the taxonomic data were then assigned to each representative sequence using the Ribosomal Database Project (RDP) classifier. The GenBank accession numbers of these OTU nucleotide sequences were MH196573-MH196855. To estimate alpha diversity, the OTU table was rarified, and the following four metrics were calculated: the Chao1 metric to estimate the richness, the observed OTU metric as the count of unique OTUs found in the sample, Shannon index, and Simpson index. In addition, the online software LefSe was utilized to select and demonstrate differentially abundant taxonomy based on the Kruskal–Wallis test and the linear discriminant analysis (LDA) score in Experiment One (Appendix A).

### 2.7. Data Analysis

One-way ANOVA was performed using GLM of SPSS version 18.0 (Statistical Package for Social Science, SPSS Inc.; Chicago, IL, USA) for feed intake, body weight, morbidity statistics, serum indices (UA, Cr, UN, and XOD), histological measurements (villus height, crypt depth and V/C ratios), alpha diversity evaluation (the observed species, Shannon, Simpson, and Chao1 indices), and bacterial abundance comparisons. The statistical unit of all the indices was the animal (n = 6 goslings/treatment) in both experiments. Statistical analyses of histological measurement data were performed based on the average of 32 measurements per gosling (four slices per gosling and eight measurements per slice). The differences between treatments were tested and the significance was accepted at *P* < 0.05.

## 3. Results

### 3.1. Kidney Injury and Gut Dysbiosis in High-Protein Diet-Induced Gosling Gout

The mortality due to gout occurred mostly at the age of 10–15 days. We found a dose-dependent relationship between the morbidity of gosling gout and dietary protein levels (Figure 1a). The morbidity in the 220CP group was 37.5%, which was considerably higher than that in the other diet groups, whereas that in the 200CP group was 7.7%. The kidneys from the dead goslings exhibited predominant gross lesions, that were pale, mottled, and swollen. The renal tubules and ureters were distended with excessive urates. Histology showed that the damage to the kidneys became severe with an increase in the dietary protein level on day 15 (Figure 1b). In particular, the morphology revealed that the renal tubular epithelial cells exhibited severe hydropic degeneration in the 220CP group.

The concentration of serum UA, Cr, UN, and the activity of XOD were also measured to evaluate renal function (Table 3). The serum UA level in goslings in the 220CP group was higher than that of the 180CP group (*P* < 0.05) on day 10, whereas the other indices (including the serum Cr and UN concentrations and the XOD activity) showed no difference among groups (*P* > 0.05). On day 15, the concentration of serum UA, Cr, and UN, and the activity of XOD gradually increased with the increase in the level of dietary protein. The serum UA concentration in the 200CP group increased compared to that in the 160CP group (*P* < 0.05). The UA concentration in the 220CP group was increased compared with that in the 160CP group (*P* < 0.01), whereas it was higher than that in 180CP group (*P* < 0.05). The Cr concentration in the 220CP group was increased compared with that in the 160CP and 180CP groups (*P* < 0.05). The UN concentration in the 160CP group was lower than that in the other groups (*P* < 0.05). The activity of XOD in the 220CP group was also increased when compared with that in the 160CP group (*P* < 0.05).

The morphological changes in the caecal tissue of goslings on diets of different protein levels were observed under a microscope (Figure 1c). With an increase in the protein level, the cecum villus height significantly decreased. Compared with that in the control, the villus height in the other groups was shorter (*P* < 0.05), especially in the 200CP and 220CP groups (*P <* 0.01). Moreover, the low villus height decreased the V/C ratio (villus height/crypt depth ratio) in the 200CP and 220CP groups (*P* < 0.01). However, there was no morphological difference in the caecal villi between goslings in the 200CP and 220CP groups (*P* > 0.05).

In the analysis of caecal microbiota by 16S rRNA gene sequencing, after quality control and chimera removal, 836,880 valid tags were retained, with an average of 34,870 tags per sample, and identified as bacterial origin. These sequences were assigned to 308 OTUs of bacterial species, based on a 97% similarity cut-off. In comparison to the control group, the observed species diversity in the high-protein diet (220CP) group was lower (*P* < 0.05) (Figure 2a,b). Species richness, evenness, and rarity are the key components of biodiversity, and are usually measured by Shannon, Simpson, and Chao1 indices, which showed no differences among goslings on different dietary protein levels (Figure 2b). After the analyses of microbiota composition, we found that the dominant phyla of these groups were *Firmicutes*, Verrucomicrobia, Bacteroidetes, and *Proteobacteria* (Figure 2c). The dominant species included *Akkermansia_unclassified*, *Bacteroides_uniformis*, *Lachnospiraceae_unclassified*, and *Bacteroides_unclassified* (Figure 2d). At the genus level, the average abundance of *Akkermansia*, *Lactococcus*, and *Butyricicoccus* decreased with the increase in protein level, whereas the abundance of *Enterococcus* increased. The average abundance of *Lactococcus* in the 160CP group was higher than that in the 220CP (*P* < 0.01) and 200CP groups (*P* < 0.05). Similarly, the samples of the 180CP group also exhibited a higher average abundance of *Lactococcus* when compared to the 220CP group (*P* < 0.05). Furthermore, the average abundance of *Butyricicoccus* in the 160CP group was higher than in the 220CP (*P* < 0.05) and 200CP groups (*P* < 0.01). Conversely, the average abundance of *Enterococcus* in the 160CP group was the lowest, but in the 220CP group, its abundance was higher (*P* < 0.05) (Figure 2e).

To explore the relationship between renal injury and gut microbiota in goslings, we analyzed the association between the average abundance of the abundant genera and the serum concentration of UA, Cr, and UN (Figure 2f). The results revealed a positive correlation between the average abundance of *Enterococcus* and *Bacteroides*, and the concentration of serum UA, whereas *Lactococcus*, *Butyricicoccus*, *Clostridium XlVb*, and *Ruminococcaceae* exhibited a negative correlation. Moreover, the average abundance of *Enterococcus* correlated positively with the serum concentration of Cr and UN, whereas the abundance of *Lactococcus* and *Butyricicoccus* negatively correlated with the serum UN level. 

### 3.2. Effects of Fermented Feed on Gut Dysbiosis and Kidney Injury Caused by High-Protein Diets

The addition of fermented feed decreased the morbidity of gosling gout caused by the high-protein diets (Figure 3a). The morbidity of gosling gout in high-protein groups was considerably higher than in the control groups (*P* < 0.01), whereas the morbidity in the high-protein groups decreased from 41.1% (HP) to 33.3% (HPF) after adding fermented feed (*P* < 0.01). The HE histology showed that the damage to the kidneys became severe in the HP and HPF groups, revealing severely hydropic degeneration in the renal tubular epithelial cells when compared to the control (Figure 3b). Fermented feed ameliorated symptoms of hydropic degeneration in the kidney when comparing the HP and HPF groups.

We measured the concentrations of serum UA, Cr, and UN, and the activity of XOD for the evaluation of renal function (Table 4). On day 7, the high-protein diets (HP) and fermented feed (F) showed significant effects on the serum UA level (*P* < 0.01) and the XOD activity (*P* < 0.05). Using fermented feed was able to relieve hyperuricemia caused by the high-protein diets. On day 14, the high-protein diets increased the UA levels (*P* < 0.05), which could be markedly reversed by fermented feed (*P* < 0.01). Meanwhile, high-protein diets increased the serum Cr concentration (*P* < 0.05), while fermented feed decreased the UN concentration (*P* < 0.01). On day 21, the effects of the high-protein diet on kidney functions were less notable, but fermented feed seemed to be helpful for kidney health. The concentration of serum Cr and the activity of XOD in groups using fermented feed were much lower than that in groups without fermented feed (*P* < 0.01), while the concentrations of serum UA (*P* =0.08) and UN (*P* =0.07) showed decreasing trends. On day 35, the high-protein diets influenced renal function, and the concentrations of serum UA, Cr, and UN all increased in the HP and HPF groups (*P* < 0.05). Using fermented feed decreased the Cr concentrations at a significant level on day 35.

We focused on the morphological changes in the caecal tissue in goslings in different groups (Figure 3c). Compared to the control (CN), the villus height in the HP groups was shorter (*P* < 0.05), but no morphological differences in crypt depth or VCR were found in goslings among these groups (*P* > 0.05).

Moreover, the caecal microbiota of goslings was analyzed by 16S rRNA gene sequencing. After quality control and chimera removal, 1,403,080 valid tags were retained, with an average of 29,852 tags per sample, and these were identified by bacterial origin. These sequences were assigned to 1889 OTUs of bacterial species, based on a 97% similarity cut-off. In the analysis of alpha diversity, the observed OTU in the CNF group increased (*P* < 0.05) in comparison to the other groups (Figure 4a,b), as did the Chao1 index (*P* < 0.05). Shannon and Simpson indices were similar among diets. Similar to Experiment One, the dominant phyla in the gosling ceca in Experiment Two were *Firmicutes*, Verrucomicrobia, Bacteroidetes, and *Proteobacteria* (Figure 4c). The dominant genera included *Akkermansia*, *Lachnospiraceae_unclassified*, *Ruminococcus2*, *Ruminococcaceae _unclassified*, *Bacteroides*, and *Escherichia* (Figure 4d). At the genus level, we compared the average abundance of *Akkermansia*, *Lactobacillus*, *Butyricicoccus*, *Bacteroides*, *Ruminococcus*, and *Enterococcus* among goslings in different groups (Figure 4e). The samples of the high-protein diet groups exhibited a higher average abundance of *Enterococcus* when compared to the other groups (*P* < 0.05). When using fermented feed in diets, the average abundance of *Lactobacillus* in the CNF and HPF groups were increased obviously (*P* < 0.05). Meanwhile, the average abundance of *Akkermansia*, *Butyricicoccus*, *Bacteroides*, and *Ruminococcus* was similar among groups (*P* > 0.05).

## 4. Discussion

Young goslings belong to a high-risk group for visceral gout. They not only lack urate oxidase, which oxidizes the poorly soluble UA to water soluble allantoin [22], resulting in the elevation of blood UA, but also have an immature kidney that is prone to injury [2,22,23]. When fed high-protein diets, the synthesis of UA increases significantly, leading to hyperuricemia or gout [24]. In Experiment One, with an increase in dietary protein level, we found that serum UA concentration and the morbidity of gosling gout increased significantly, especially when the CP level was increased to 220 g/kg. These results concur with those of previous studies in chickens [22,25]. Meanwhile, high-protein diets also adversely affected renal function in the development of gosling gout. The serum Cr and UN concentrations (parameters used to assess overall kidney function) in the 220CP treatment group were higher than those in the 160CP group on day 15. This might be due to the impairment of renal function, such as glomerular filtration [26,27]. Xanthine oxidase, a member of the molybdenum hydroxylase flavoprotein family, plays a vital role in the conversion of xanthine to UA, and therefore the increase in XOD activity can cause the hyperplasia of free radicals [28]. Most kidney function damage associated with hyperuricemia is not caused directly by UA, but by superoxide-free radicals produced during the formation of UA by XOD [28]. In the present study, we found an increase in the activity of XOD in high-protein diet groups with increasing in the dietary protein level on day 15. This indicates that the kidneys of goslings on high-protein diets might be under an additional attack by free radicals. Although some scientists believe that the consumption of high protein is not necessarily related to renal injury because of the self-adjusting ability of the kidney [14,22,29], the kidneys of young goslings are not fully developed and therefore might be more easily damaged by high purine levels. High protein-induced physiological consequences might produce a significant UA load in the kidneys, increasing the risk of kidney disease, especially in individuals with pre-existing kidney dysfunction or mild renal insufficiency [21,30,31]. Thus, the high-protein diets could lead to gout by inducing kidney damage in goslings. 

The effects of diet on kidney health are largely dependent on the metabolic activities of the intestinal microbiota [32,33]. The microbiota compositions in the cecum of goslings changed because of the high-protein diets in the present study. At the phylum level, the results indicated that *Bacteroides* was high in the gout goslings, and this exhibited a positive correlation with the serum concentration of serum UA. Indeed, a diet rich in protein favors the growth of members of *Bacteroidetes* [34]. Although *Bacteroides*, as the dominant phyla in geese [35,36], is not a common pathogen, many studies have reported that its proliferation is closely related to gout in humans [14,22,37], reflecting a close relationship between the gut microbiota and kidney health. At the genus level, we found that the average abundance of *Lactococcus* and *Butyricicoccus* decreased with increase in the protein levels, whereas the average abundance of *Enterococcus* increased. Meanwhile, the average abundance of *Lactococcus* and *Butyricicoccus* exhibited a negative correlation with the serum concentration of UA and UN, whereas *Enterococcus* exhibited a positive correlation. This indicates a decrease in beneficial bacteria, but an increase in pathogenic bacteria accompanied by renal dysfunction, which is consistent with studies in humans with gout, multiple kidney stones, or diabetic nephropathy [14,37]. Usually, these beneficial bacteria maintain adequate gut barrier function to avoid intestinal bacteria being translocated into liver, kidney, and mesenteric lymphatic vessels [38], which might decrease the occurrence of gosling gout [7]. Even, some short-chain fatty acids, including lactic acid produced by *Lactococcus* and butyric acid produced by *Butyricicoccus*, can directly inhibit the proliferation of glomerular mesangial cells induced by LPS, and then reverse the production of reactive oxygen species to protect the kidneys [39]. Conversely, *Enterococcus*, the most common source of human infection according to epidemiological investigations [40], increased significantly in the high-protein groups, disrupting the intestinal microecological balance and potentially transferring bacterial infections from the gut to the kidney.

Some studies have addressed the influence of protein-rich diets not only on microbiota composition, but also on the intestinal histomorphology [41]. In the present study, we found that the villus height decreased with an increase in dietary protein level and the V/C ratio changed correspondingly, and that high-protein diets could change the morphology of the epithelial cells in goslings. An increase in dietary protein intake should accordingly prompt an increase in the unabsorbed proteins, peptides, and amino acids entering the cecum. These substances will be degraded by endogenous as well as microbiota-produced proteases into peptides and free amino acids. Some of the products are precursors of numerous bacterial metabolites, including ammonia, hydrogen sulfide, amines, indoles, phenols, and organic acids, and have cytotoxic effects on the intestinal epithelial cells in the distal intestinal tract [11,33,42]. Although proteolytic fermentation produces some compounds beneficial for the gut microbiota, intestinal putrefaction is considered more detrimental to the health of goslings than other organisms [7,43,44]. Overall, we believe that the high-protein diets decreased the growth of beneficial bacteria, but promoted the growth of pathogenic bacteria in the cecum of goslings, damaging the health of the intestinal epithelium, and therefore potentially affecting renal function in gout development.

In recent decades, the fermentation process has been employed to produce functional feeds that exhibit beneficial influences on gut ecosystems and morphology, immune function, as well as growth performance of birds [19]. In Experiment Two, we found that using fermented feed could alleviate the occurrence of gout caused by the high-protein diets and improve renal function in goslings. The cumulative incidence of gout decreased in the HPF group compared to that in the HP group, as did the concentration of serum UA. Although previous studies have shown that fermentation increased the CP content of feed ingredients [45], which might work against the prevention for gout, we found that using only 5% fermented feed had limited effects on enhancing the CP levels in this study. More importantly, we believe that the positive influence of fermented feed on renal function in goslings came from its regulation of gut microbiota. In general, fermented products tend to have higher and lower number of LAB (mainly *Lactobacillus* and *Lactococcus*) and *Enterobacteriaceae*, respectively, maintaining healthy gut microbiota and strengthening the immune system [16]. In a previous study, we found that fermented feeds had beneficial effects on the microflora composition of adult geese, affecting intestinal health and production performance [15]. In the present study, the gut microbiota and villus morphology in goslings fed fermented feed was improved in the high-protein groups. We found that fermented feed made *Lactobacillus* increase and *Enterococcus* decrease in the cecum of goslings. These properties make fermented feeds particularly beneficial for healthy gastrointestinal function and well-being in birds [17,46]. Moreover, *Lactobacillus* can even efficiently decompose purine nucleotides directly (both in vitro and in vivo), and is an effective treatment for hyperuricemia in rats [47]. This could relieve, and even reverse, the changes in the gut or kidney brought about by the high-protein diets, so we suspected this would be one of the most important reasons why fermented feed alleviates gout caused by high-protein diets.

In addition, using fermented feeds prevented gosling gout better than the direct addition of a complex probiotic preparation (mainly including *Lactobacillus plantarum*, *Bacillus subtilis*, and *Bacillus coagulans*) in a previous study (unpublished data). Thus, besides the probiotics, some other metabolites produced in the fermentation process including lactic acid, enzymes, and organic acids [15,45] might also play important roles in gut or kidney protection, reducing the occurrence of gosling gout; however, more investigation on this needs to be done. Using fermented feed is an alternative strategy in the control and prevention of gosling gout.

## 5. Conclusions

In conclusion, we found that high-protein diets induced kidney damage and promoted the occurrence of gout in goslings. High-protein diets injured intestinal epithelial cells and suppressed the growth of beneficial bacteria but also promoted the growth of pathogenic bacteria in the cecum, thereby affecting renal function. Using fermented feed alleviates the gout occurrence caused by the high-protein diets and improves renal function, probably because of its regulations of caecal microbiota. Fermented feed increased *Lactobacillus* and decreased *Enterococcus* in the ceca of goslings. Given all this, we suggest that goose farmers should strictly control the proportion of protein in gosling feed and consider using fermented feed as an effective measure to control the occurrence of gosling gout.

## Figures and Tables

**Figure 1 animals-10-02139-f001:**
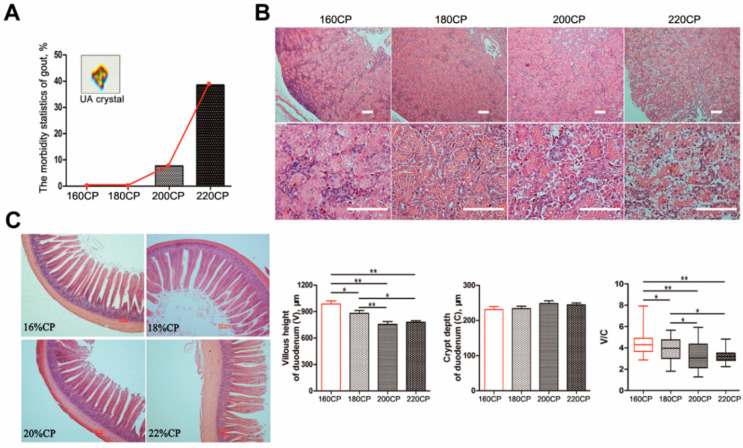
Evaluation of changes in renal injury and intestinal morphology in goslings with different protein diets. (**A**) The morbidity of gout in goslings with different protein diets; (**B**) morphological observation of the kidney by HE stain (Bar = 100 μm). (**C**) Morphological observation and of the intestinal villus among groups (HE staining, Bar = 200 μm); (**D**) the comparisons of villus height, crypt depth, and their ratio in duodenum. Asterisks represent differences between groups being significant (* *P* < 0.05, ** *P* < 0.01).

**Figure 2 animals-10-02139-f002:**
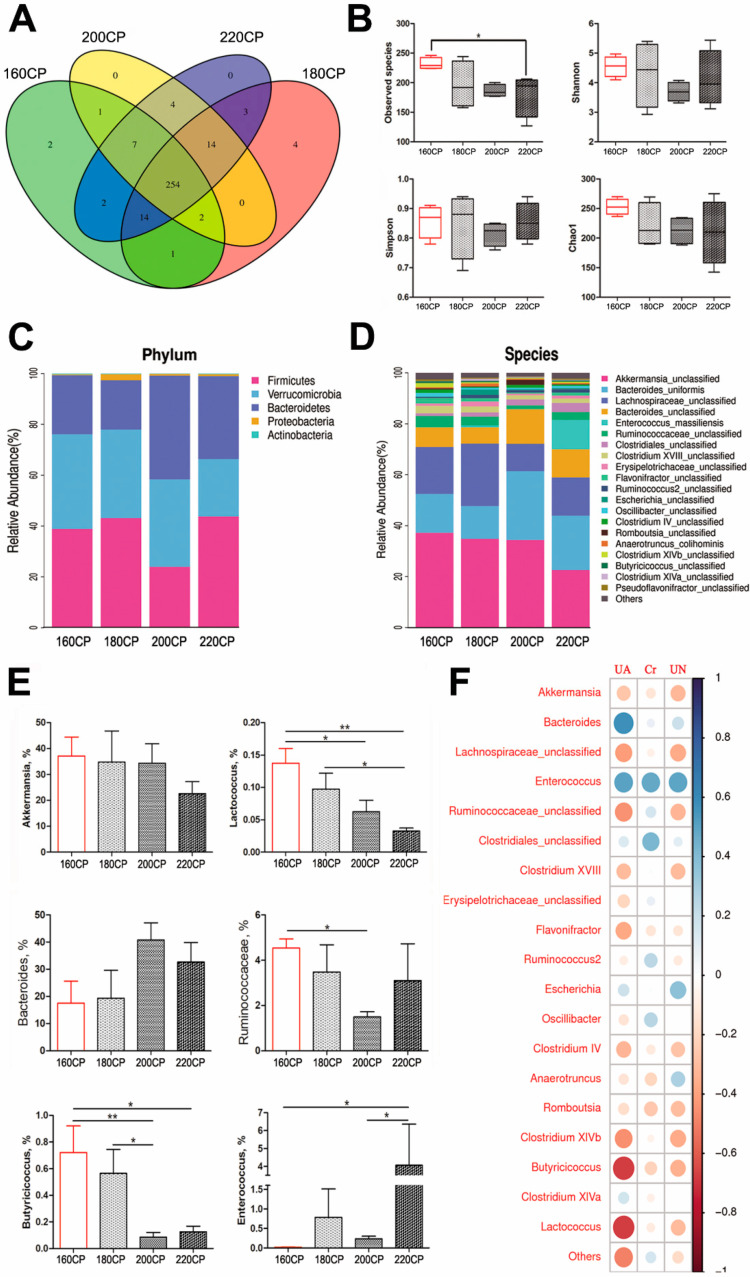
The degrees of gut dysbiosis in cecum of goslings with different protein diets. (**A**) Venn diagram of the observed species; (**B**) total observed species and alpha diversity indexes, n = 6; (**C**,**D**) microbiome composition at the phylum and species levels, n = 6; (**E**) the average abundances of some genera among groups, n = 6; (**F**) correlation analysis between the abundant genera and the serum concentration of UA, Cr, and BUN in goslings with different protein diets by the R program. The circle size and color intensity represent the magnitude of correlation. Blue circle = positive correlation; red circle = negative correlation. UA: uric acid, Cr: creatinine, BUN: blood urea nitrogen. Asterisks represent differences between groups being significant (* *P* < 0.05, ** *P* < 0.01).

**Figure 3 animals-10-02139-f003:**
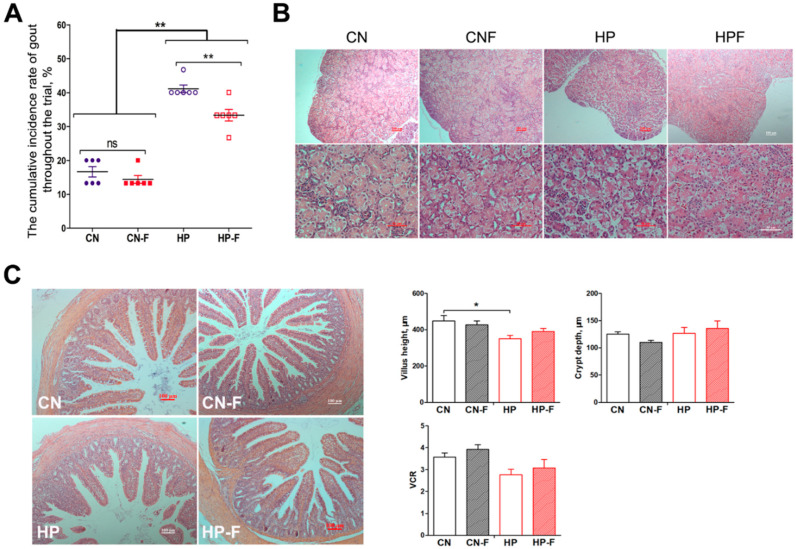
Evaluation of changes in renal injury and intestinal morphology in goslings among groups (groups: CN: a basal diet; CNF: replaced 5.0% basal diet with fermented feed; HP: a high-protein diet; HPF: replaced 5.0% high-protein diet with fermented feed). (**A**) The morbidity of gout in goslings among groups; (**B**) morphological observation of the kidney by HE stain (Bar = 100 μm). (**C**) Morphological observation and of the intestinal villus among groups (HE staining, Bar = 200 μm); (**D**) the comparisons of villus height, crypt depth, and their ratio in cecum. CN: 162.9 g/kg CP, a basal diet; CNF: 167.5 g/kg CP, replacing 5.0% basal diet with fermented feed; HP: 229.7 g/kg CP, a high-protein diet; HPF: 230.7 g/kg CP, replacing 5.0% high-protein diet with fermented feed. Asterisks represent differences between groups being significant (* *P* < 0.05, ** *P* < 0.01).

**Figure 4 animals-10-02139-f004:**
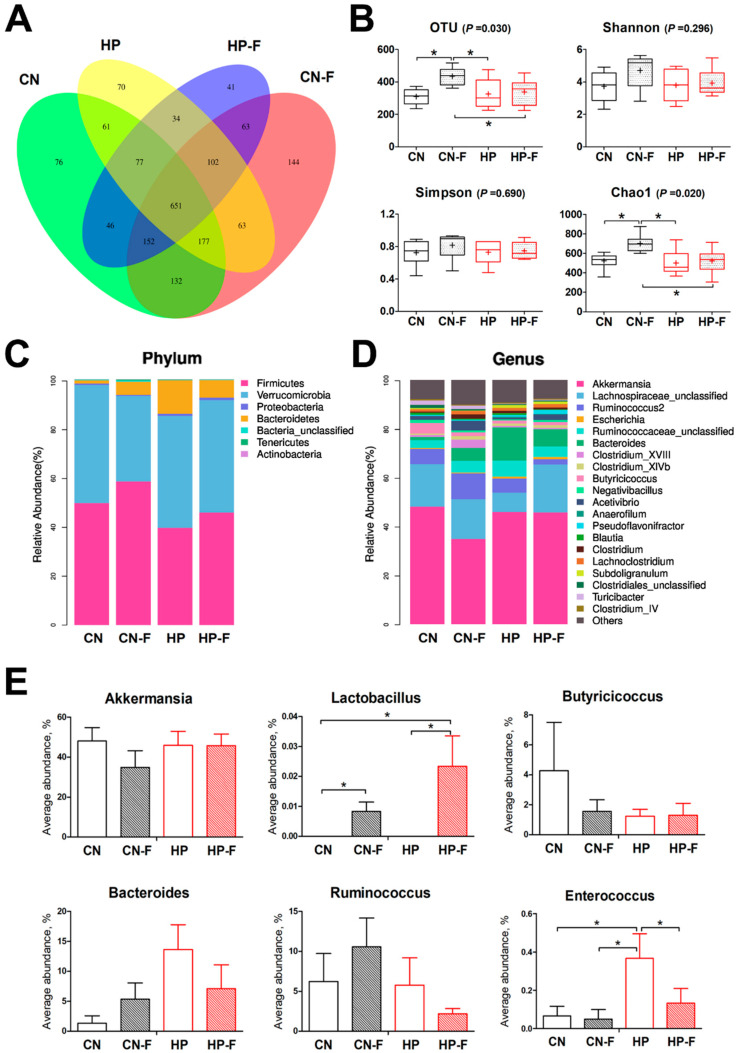
The degrees of gut dysbiosis in cecum of goslings among groups (groups: CN: a basal diet; CNF: replaced 5.0% basal diet with fermented feed; HP: a high-protein diet; HPF: replaced 5.0% high-protein diet with fermented feed). (**A**) Venn diagram of the observed OTUs; (**B**) total OTUs and alpha diversity indexes, n = 6; (**C**,**D**) microbiome composition at the phylum and genus levels, n = 6; (**E**) the average abundances of some genera among groups, n = 6. CN: 162.9 g/kg CP, a basal diet; CNF: 167.5 g/kg CP, replacing 5.0% basal diet with fermented feed; HP: 229.7 g/kg CP, a high-protein diet; HPF: 230.7 g/kg CP, replacing 5.0% high-protein diet with fermented feed. Asterisks represent differences between groups being significant (* *P* < 0.05).

**Table 1 animals-10-02139-t001:** Composition of experimental diets in Experiment One.

	CP Levels			
	160CP	180CP	200CP	220CP
Feed compositions				
Corn, g/kg	523	496	491	468
Bean pulp, g/kg	200	214	255	300
Fish meal, g/kg	−	39	59	77
Bran, g/kg	78	79	62	38
Rice bran, g/kg	156	124	80	62
Rice hulls, g/kg	2	7	12	15
CaCO_3_, g/kg	15	15	15	15
CaHPO_4_, g/kg	13	13	13	12
NaCl, g/kg	3	3	3	3
Vitamin/mineral premix ^1^, g/kg	10	10	10	10
Nutritional ingredients ^2^				
Poultry ME, Kcal/kg	2700	2700	2700	2700
CP, g/kg	160	180	200	220
CF, g/kg	38	38	38	38
Calcium, g/kg	10	10	10	10
Phosphorus, g/kg	4	4	4	4

ME: Metabolizable energy, CP: crude protein, CF: crude fiber. ^1^ Provided per kilogram of diet: vitamin D, 1000 IU; vitamin A, 4500 IU; vitamin E, 30 IU; vitamin K3, 1.3 mg; vitamin B1, 2.2 mg; vitamin B2, 10 mg; vitamin B12, 1.013 mg; vitamin B6, 4 mg; calcium, 7.5 mg; niacin, 20 mg; folic acid, 0.5 mg; biotin, 0.04 mg; copper, 7.5 mg; iron, 60 mg; zinc, 65 mg; manganese, 110 mg; iodine, 1.1 mg; selenium, 0.15 mg. ^2^ The ME was calculated value, and the CP, CF, calcium, and phosphorus were analyzed values. The analysis of chemical compositions of feedstuffs was entrusted to the Suqian Kangdifuer Feed Limited Company (Suqian, China).

**Table 2 animals-10-02139-t002:** Composition of experimental diets (1–35 days) in Experiment Two.

	CN ^2^	CNF	HP	HPF
Feed compositions				
Corn, g/kg	589	589	460	460
Bean pulp, g/kg	160	160	240	240
Corn gluten meal, g/kg	40	40	100	100
Fish meal, g/kg	0	0	50	50
Bran, g/kg	110	110	69	69
Rice bran, g/kg	30	30	30	30
Rice hulls, g/kg	30	30	20	20
CaCO_3_, g/kg	11	11	11	11
CaHPO_4_, g/kg	13	13	3	3
NaCl, g/kg	3	3	3	3
Vitamin/mineral premix ^1^, g/kg	10	10	10	10
Basal diet supplemented with 5.0% fermented feed	−	+	−	+
Nutritional ingredients				
Poultry ME, KJ/kg	11.2	11.3	11.4	11.4
CP, g/kg	162.9	167.5	229.7	230.7
CF, g/kg	41.3	38.2	39.5	33.8
Crude Fat, g/kg	20.7	19.7	27.3	19.5
Calcium, g/kg	11.6	11.2	11.9	11.9
Phosphorus, g/kg	6	6.1	7.2	7.4

ME: Metabolizable energy, CP: crude protein, CF: crude fiber. ^1^ Provided per kilogram of diet: vitamin D, 1000 IU; vitamin A, 4500IU; vitamin E, 30 IU; vitamin K3, 1.3 mg; vitamin B1, 2.2 mg; vitamin B2, 10 mg; vitamin B12, 1.013 mg; vitamin B6, 4 mg; calcium, 7.5 mg; niacin, 20 mg; folic acid, 0.5 mg; biotin, 0.04 mg; copper, 7.5 mg; iron, 60 mg; zinc, 65 mg; manganese, 110 mg; iodine, 1.1 mg; selenium, 0.15 mg. ^2^ CN: 162.9 g/kg CP, a basal diet; CNF: 167.5 g/kg CP, replacing 5.0% basal diet with fermented feed; HP: 229.7 g/kg CP, a high-protein diet; HPF: 230.7 g/kg CP, replacing 5.0% high-protein diet with fermented feed.

**Table 3 animals-10-02139-t003:** Effects of protein levels on kidney function in goslings in Experiment One.

Indexes	Groups				SEM	*p*-Value
	160CP	180CP	200CP	220CP		
Kidney function, Day 10 ^1^						
UA, μmol·L^−1^	94.57 ^ab^	72.83 ^a^	106.52 ^ab^	210.87 ^b^	43.06	0.04
Cr, μmol·L^−1^	17.70	19.65	18.26	19.95	1.30	0.57
UN, mmol·L^−1^	0.89	2.05	1.28	1.53	0.29	0.09
XOD, U·L^−1^	4.84	4.79	4.18	5.19	0.39	0.36
Kidney function, Day 15 ^1^						
UA, μmol·L^−1^	248.50 ^Aa^	294.72 ^ABab^	350.99 ^ABbc^	400.22 ^Bcd^	31.48	< 0.01
Cr, μmol·L^−1^	10.37 ^a^	11.16 ^a^	13.67 ^ab^	18.68 ^b^	2.28	0.04
UN, mmol·L^−1^	0.38 ^a^	0.91 ^b^	1.01 ^b^	1.04 ^b^	0.17	0.02
XOD, U·L^−1^	4.42 ^a^	5.15 ^ab^	5.62 ^ab^	6.28 ^b^	0.80	0.03

CP: crude protein, UA: uric acid, Cr: creatinine, UN: urea nitrogen, XOD: xanthine oxidase, SEM: square error of Mean. ^1^ n = 6 goslings/treatment. ^a,b,c,d,^ Values within a row with different superscripts differ significantly at *p*-value < 0.05. ^A,B^ Values within a row with different superscripts differ significantly at *p*-value < 0.01.

**Table 4 animals-10-02139-t004:** Effects of high-protein diet and fermented feed on kidney function in goslings in Experiment Two.

Indexes	Groups ^1^				SEM	*p*-Value		
	CN	CNF	HP	HPF		HP	F	HP*F
Day 7								
UA, μmol·L^−1^	407.12	276.12	578.38	392.25	28.96	0.004	0.002	0.548
Cr, μmol·L^−1^	18.89	22.01	19.24	16.70	1.33	0.367	0.917	0.303
UN, mmol·L^−1^	1.54	1.96	1.83	1.53	0.09	0.697	0.740	0.143
XOD, U·L^−1^	6.09	6.92	10.71	6.17	0.55	0.040	0.048	0.106
Day 14								
UA, μmol·L^−1^	289.25	151.38	444.75	251.12	30.94	0.019	0.003	0.592
Cr, μmol·L^−1^	10.10	9.40	14.13	10.47	0.77	0.038	0.064	0.762
UN, mmol·L^−1^	1.62	1.18	1.90	1.02	0.10	0.723	0.001	0.200
XOD, U·L^−1^	6.99	6.95	8.88	6.85	0.31	0.853	0.092	0.103
Day 21								
UA, μmol·L^−1^	202.88	137.62	247.25	147.75	22.96	0.553	0.080	0.708
Cr, μmol·L^−1^	7.57	7.08	14.45	5.63	0.97	0.095	0.006	0.113
UN, mmol·L^−1^	1.03	0.88	1.17	0.87	0.06	0.588	0.070	0.509
XOD, U·L^−1^	6.07	5.23	8.48	4.77	0.38	0.116	0.001	0.124
Day 35								
UA, μmol·L^−1^	382.83	293.00	813.00	780.83	55.24	0.000	0.299	0.620
Cr, μmol·L^−1^	11.34	5.84	21.44	13.43	1.46	0.000	0.002	0.509
UN, mmol·L^−1^	1.15	1.00	1.80	1.24	0.11	0.035	0.083	0.307
XOD, U·L^−1^	5.91	7.66	5.66	5.25	0.44	0.127	0.431	0.211

HP: high-protein diet, F: fermented feed, UA: uric acid, Cr: creatinine, UN: urea nitrogen, XOD: xanthine oxidase, SEM: square error of Mean. CN: 162.9 g/kg CP, a basal diet; CNF: 167.5 g/kg CP, replacing 5.0% basal diet with fermented feed; HP: 229.7 g/kg CP, a high-protein diet; HPF: 230.7 g/kg CP, replacing 5.0% high-protein diet with fermented feed; HP*F: the interaction effects between HP (high-protein diet) and F (fermented feed) dynamics variables. ^1^ n = 6 repeats/treatment.

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
