# Peer review of "Fermented Feed Supplement Relieves Caecal Microbiota Dysbiosis and Kidney Injury Caused by High-Protein Diet in the Development of Gosling Gout"

_animals, 2020, doi:10.3390/ani10112139_

Round 1
Reviewer 1 Report
Please see the attached file for details

Author Response
Thank you for your letter and for your comments concerning our manuscript entitled “Fermented feed supplement relieves gut microbiota dysbiosis and kidney injury caused by high-protein diet in the development of gosling gout” ID: 959994. Those comments are all valuable and very helpful for revising and improving our paper as well as the important guiding significance to our research. We have studied comments carefully and have made correction which we hope meet with approval. Revised portion are marked in yellow in the paper.
Line 21: Allocated instead of allotted
Response: We have changed “allotted” to “allocated”. (line 21)
Line 26: Why “then” need to be bold?
Response: We have changed the format. The “then” need not to be bold. (line 27)
Line 47: What is CP? Crude protein? Please write the full form of this term and the
abbreviation in the bracket.
Response: We have written the full form of crude protein and the abbreviation in the bracket. (line 52)
Line 47: fishmeal instead of fish meal
Response: We have changed “fish meal” to “fishmeal”. (line 53)
Line 58: microbial compositions?
Response: The compositions of large intestine contents, epithelial cell morphology and bacterial metabolism. (line 62)
Line 63: What is the LPS? Add the full form in the text please.
Response: We have written the full form of lipopolysaccharide and the abbreviation in the bracket. (line 68)
Line 75: The full form should be added for IgG
Response: We have written the full form of immunoglobulin g and the abbreviation in the bracket. (line 80)
Line 66, 80: “Gut microbiota” need to be used instead of “intestinal microecology” or “intestinal flora” or “intestinal microbiomes” throughout the manuscripts.
Response: We have changed “intestinal microecology” or “intestinal flora” or “intestinal microbiomes” to “gut microbiota” (lines 64, 71, 85 …)
Line 107: Feed intake and uneaten instead of “feed offered and refused”
Response: We have changed “feed offered and refused” to “feed intake and uneaten”. (line 112)
Line 91-119: It is not clear about how long the experimental period? 15 days? This section needs to be reconstruction and rephrased.
Response: The experimental period was 15 days. This section had been reconstruction and rephrased. (line 109)
Line 143: The full form of CF should be mentioned first.
Response: We had added the full form of CF. (line 154)
Line 162: Table 3 and 4?
Response: Yes, it was. We have changed it to “Table 3 and 4”. (line 173)
Line 168: Intra-assay?
Response: Yes, it was. We have changed it to “Intra-assay”. (line 179)
Line 207: The full form of LDA should be mentioned first.
Response: We have written the full form of linear discriminant analysis and the abbreviation in the bracket. (line 219)
Reviewer 2 Report
The research article by Xi et al. focused on the effects of gut microbiota dysbiosis and kidney injury affected by hight protein diet. The authors have employed suitable metodologies and discussed they findings well.I only recommended to correct some part in which they refer to intestinal microbiota. If they want to use this term, they should be able to also provide data on duodenum, jeiunum and ileum. Based on my experience, dietary treatments influence mostly the ileum tract under different aspects, microbiota, morphology, and pH. Moreover the authors should very brifly describe the feed fermentation procedure with times,temperatures and pH. I suggest to highlighting more the positive effects on kidney than the morphology and microbiota, moreover, I suggest analysing in deep the three different part of intestine for further studies.
Author Response
Thank you for your letter and for your comments concerning our manuscript entitled “Fermented feed supplement relieves gut microbiota dysbiosis and kidney injury caused by high-protein diet in the development of gosling gout” (ID: 959994). Those comments are all valuable and very helpful for revising and improving our paper as well as the important guiding significance to our research. We have studied comments carefully and have made correction which we hope meet with approval.
Firstly, we feel very sorry for the inaccurate description of “intestinal microbiota”, and used “caecal microbiota” instead of it in some parts of the paper (title, abstract, key words, method…). Then, we brifly described the feed fermentation procedure with times, temperatures and pH in the “method” section, and revised portion were marked in yellow in the paper. “For feed fermentations, a seed sourdough was prepared with fermentation substrate (a mixture of corn, soybean meal and bran with certain proportion) and multiple-strain culture (Lactobacillus plantarum, Bacillus subtilis, saccharomycetes and so on). The seed sourdough was used to inoculate feed fermentation, and the fermentation material needed to be stacked to 60-80 cm high in the fermenter. During fermentation, the fermentation material needed to be stirred, ventilated and refrigerated, when the temperature of the fermented material in the fermenter was greater than 45℃. After 72-120 h of fermentation, the pH of fermentation material reached 3.8-4.2, denoting the completion of fermentation.” Finally, thank you for your suggestion. Indeed, more attention should be paid to the positive effects on kidney, as well as on the other three different parts of intestine. There's more work needed to be done.
Thanks again!
Reviewer 3 Report
Fermented feed supplement relieves gut microbiota dysbiosis and kidney injury caused by high-protein diet in the development of gosling gout
The paper presents some interesting results of beneficial effect fermented feed supplement relieves gut microbiota dysbiosis and kidney injury caused by high-protein diet in the development of gosling gout. The research was performed with high precision, the theme of the work very topical from the point of view of bird welfare and food safety.
Author Response
Thank you for your letter and for your comments concerning our manuscript entitled “Fermented feed supplement relieves gut microbiota dysbiosis and kidney injury caused by high-protein diet in the development of gosling gout”.
Reviewer 4 Report
This paper explored if fermented feed supplements relieved gut microbiota dysbiosis and kidney injury in goslings offered high protein diets. The conclusions are supported by the findings and it is a neat study.
I have only a few comments;
It would be helpful to add some explanation in the introduction as to why goslings are fed high protein diets.
Extensive editing of the standard of English is required throughout. For example, in some areas the sentence structure is odd and I'm unsure what it means; Line 24: "While, the beneficial microbes, including Akkermansia, lactococcus, and Butyricicoccus were enriched in the ceca of healthy goslings, while the microbes Enterococcus, Enterobacteriaceae, and Bacteroides were enriched in those with gout."
In table footnotes, please also define CN and CNF (for example, Line 319).
Author Response
Thank you for your letter and for your comments concerning our manuscript entitled “Fermented feed supplement relieves gut microbiota dysbiosis and kidney injury caused by high-protein diet in the development of gosling gout” ID: 959994.
Those comments are all valuable and very helpful for revising and improving our paper as well as the important guiding significance to our research. We have studied comments carefully and have made correction which we hope meet with approval. Revised portion are marked in yellow in the paper. We have added some explanation in the introduction as to why goslings are fed high protein diets. “During recent years, the modernisation of the goose industry in China has resulted in a growth rate-oriented production that primarily is based on feeding the geese high-protein complete feeds, instead of the traditional herbaceous roughages, but also leads to physical problems for the geese.” In addition, we also invited a professional team to editing the standard of English in the paper, and had uploaded a prove file in the attachment. The definition of CN and CNF in table footnotes are also added in the table.
